# Human PSEN1 Mutant Glia Improve Spatial Learning and Memory in Aged Mice

**DOI:** 10.3390/cells11244116

**Published:** 2022-12-18

**Authors:** Henna Jäntti, Minna Oksanen, Pinja Kettunen, Stella Manta, Lionel Mouledous, Hennariikka Koivisto, Johanna Ruuth, Kalevi Trontti, Hiramani Dhungana, Meike Keuters, Isabelle Weert, Marja Koskuvi, Iiris Hovatta, Anni-Maija Linden, Claire Rampon, Tarja Malm, Heikki Tanila, Jari Koistinaho, Taisia Rolova

**Affiliations:** 1A.I. Virtanen Institute for Molecular Sciences, University of Eastern Finland, 70211 Kuopio, Finland; 2Broad Institute, Cambridge, MA 02142, USA; 3Neuroscience Center, HILIFE, University of Helsinki, 00014 Helsinki, Finland; 4Centre de Recherches sur la Cognition Animale (CRCA), Université de Toulouse, CNRS, UPS, CEDEX 09, 31062 Toulouse, France; 5Centre de Biologie Intégrative (CBI), Université de Toulouse, CNRS, UPS, 31062 Toulouse, France; 6Institute of Clinical Medicine, University of Eastern Finland, 70211 Kuopio, Finland; 7SleepWell Research Program, Faculty of Medicine, University of Helsinki, 00014 Helsinki, Finland; 8Department of Psychology and Logopedics, University of Helsinki, 00014 Helsinki, Finland; 9Department of Physiology and Pharmacology, Karolinska Institutet, 17165 Solna, Sweden; 10Department of Pharmacology, Faculty of Medicine, University of Helsinki, 00014 Helsinki, Finland

**Keywords:** Alzheimer’s disease, presenilin, iPSC, experimental transplantation, oligodendrocyte precursor cells, astrocytes

## Abstract

The *PSEN1* ΔE9 mutation causes a familial form of Alzheimer’s disease (AD) by shifting the processing of amyloid precursor protein (APP) towards the generation of highly amyloidogenic Aβ42 peptide. We have previously shown that the *PSEN1* ΔE9 mutation in human-induced pluripotent stem cell (iPSC)-derived astrocytes increases Aβ42 production and impairs cellular responses. Here, we injected *PSEN1* ΔE9 mutant astrosphere-derived glial progenitors into newborn mice and investigated mouse behavior at the ages of 8, 12, and 16 months. While we did not find significant behavioral changes in younger mice, spatial learning and memory were paradoxically improved in 16-month-old *PSEN1* ΔE9 glia-transplanted male mice as compared to age-matched isogenic control-transplanted animals. Memory improvement was associated with lower levels of soluble, but not insoluble, human Aβ42 in the mouse brain. We also found a decreased engraftment of *PSEN1* ΔE9 mutant cells in the cingulate cortex and significant transcriptional changes in both human and mouse genes in the hippocampus, including the extracellular matrix-related genes. Overall, the presence of *PSEN1* ΔE9 mutant glia exerted a more beneficial effect on aged mouse brain than the isogenic control human cells likely as a combination of several factors.

## 1. Introduction

Alzheimer’s disease (AD) is the most common form of dementia affecting up to 30% of individuals above the age of 85 [1]. Deletion of exon 9 in the human *PSEN1* gene encoding for presenilin 1 (*PSEN1* ΔE9) causes an early-onset form of AD characterized by the accumulation of amyloid plaques, neurofibrillary tangles, and cerebral amyloid angiopathy [2,3]. The *PSEN1* gene encodes the active site of gamma-secretase, an enzyme involved in the generation of Aβ peptides from full-length amyloid precursor protein (APP) through intramembrane cleavage. The *PSEN1* ΔE9 mutation shifts the intramembrane cleavage level resulting in a higher ratio of a more aggregation-prone Aβ42 isoform to shorter Aβ40 and Aβ38 isoforms [4,5,6,7]. In contrast to humans, the *PSEN1* ΔE9 mutation in mice does not cause an AD-like phenotype per se but enhances amyloid plaque formation caused by overexpression of a mutant form of APP [8].

Induced pluripotent stem cell (iPSC) technology allows the generation of human brain cells in culture and the studying of human-specific disease mechanisms. AD research on the early-onset form of AD has primarily focused on Aβ production in neurons. However, we have previously shown that the *PSEN1* ΔE9 mutation increases the production of Aβ42 and causes metabolic dysfunction in human iPSC-derived astrocytes in vitro as well [9,10]. Pathological changes in mutant astrocytes included impaired calcium homeostasis, increased production of reactive oxygen species, decreased production of lactate, and impaired cytokine response to inflammatory stimuli. Furthermore, the co-cultures of human iPSC-derived neurons with mutant astrocytes impaired neuronal responses to glutamate and gamma-aminobutyric acid (GABA) [9]. 

It has been previously shown that human iPSC-derived glial progenitors survive well when transplanted into the brains of neonatal immunodeficient mice giving rise to the populations of human astrocytes, oligodendrocytes, and oligodendrocyte precursors (OPCs, also known as NG2 glia) [11,12]. The purpose of this study was to investigate the effect of the *PSEN1* ΔE9 mutation on human glial phenotype in vivo, in particular, its effect on mouse behavior and cognition as well as Aβ deposition. Here, we show that while there were no significant behavior differences in younger animals, aged 16-months-old, *PSEN1* ΔE9-transplanted mice exhibited significantly improved spatial learning and memory as compared to isogenic control (CTRL)-transplanted animals, suggesting that the mutant cells had a more beneficial effect on the aged mouse brain. *PSEN1* ΔE9-transplanted mice also had lower cortical levels of soluble Aβ42 as compared to the CTRL-transplanted animals.

## 2. Materials and Methods

The experimental setup is shown in Figure 1.

### 2.1. Human iPS Cells and Mice

Two pairs of previously established *PSEN1* ΔE9 mutant and isogenic control human-derived iPSC lines [9] and two control lines from non-related human individuals [13] were used in this study (Table 1). The use of the cells was approved by the committee on Research Ethics of Northern Savo Hospital District (license nr. 123/2016).

Rag1 mice (B6.129S7-Rag1tm1Mom/J, RRID:IMSR_JAX:002216) were bought from the Jackson Laboratory (Bar Harbor, ME, USA) and maintained at the Laboratory Animal Centre, University of Eastern Finland, Kuopio, Finland (Batch I) or the University of Helsinki, Helsinki, Finland (Batch II). The mice were housed under a 12-h light/dark cycle with food and water available ad libitum. All the animal experiments were approved by the National Animal Experiment Board of Finland and followed the Council of Europe Legislation and Regulation for Animal Protection.

### 2.2. Glial Differentiation

iPSCs were grown in feeder-free conditions in Essential 8 Medium (Thermo Fisher Scientific, Waltham, MA, USA) on Matrigel (growth factor reduced; Corning, Corning, NY, USA)-coated 3.5 cm dishes. The cells were passaged with 5 mM EDTA every 4–5 days. iPSCs were differentiated to neuroepithelial cells by dual SMAD inhibition using SB431542 10 µM (Merck, Rahway, NJ, USA) and LDN-193189 200 nM (Selleck Chemicals, Houston, TX, USA) for 12 days as previously described [9,10]. Areas with rosettes were mechanically lifted and cultured in suspension on ultra-low attachment plates (Corning) to initiate sphere formation. Spheres were expanded for 6.5–7.5 months in the astrocyte sphere medium containing DMEM/F12 supplemented with 5% N2 supplement, 2 mM Glutamax, 1 × non-essential amino acids, 50 IU/mL penicillin, 50 μg/mL streptomycin (all from Thermo Fisher Scientific), 1 U/mL Heparin (LEO Pharma, Ballerup, Denmark), 10 ng/mL bFGF, and 10 ng/mL EGF (both from Peprotech, Thermo Fisher Scientific). Half of the medium was replaced every 2–3 days, and spheres were manually cut once a week with a scalpel.

### 2.3. Transplantation

The transplantation protocol was adapted from [14]. The 6.5–7.5-months-old astrospheres were dissociated with Accutase and resuspended at the concentration of 100,000 cells/µL in phosphate-buffered saline (PBS) prior to transplantation. Neonatal mouse pups (P0-P1) were anesthetized on ice until the reflexes were lost and placed into a stereotaxic apparatus (Hamilton, Bonaduz, Switzerland). A total of 200,000 cells were injected intracerebroventricularly at 4 different injection sites (M/L ± 0.8 mm; A/P 1.0 mm, 2.0 mm; D/V −1.5 mm from lambda point; 0.5 μL each) using a 33 G needle cut at 30° angle (Hamilton) at rate 3 μL/min. Further, 100,000 cells in 1 μL PBS were injected into the middle of the cerebellum by freehand. After the injections, the pups were placed in a +37 °C chamber for 5 min to recover from cold anesthesia and then returned to their mother. Mice were checked daily. After weaning at 21 days, each litter was moved to a separate enriched housing. Altogether, 48 pups were injected in Batch I and 38 pups were injected in Batch II (Table 2).

### 2.4. Sample Collection

BrdU was injected (i.p. 50 mg/kg) every day for 10 days before sacrifice. At 18 months of age, mice were terminally anesthetized with Tribromoethanol (Avertin) and perfused transcardially with ice-cold heparinized saline (2500 IU/l; Heparin LEO 5000 IU/mL, Leo Pharma A/S, Ballerup, Denmark). The brains were removed immediately and divided into two hemispheres. From one hemisphere, the hippocampus, cortex, and cerebellum were quickly dissected out, snap-frozen in liquid nitrogen, and stored at −80 °C until use. The other hemisphere was fixed in 4% formaldehyde solution in 0.1 M phosphate buffer (PB), pH 7.4, for 20–22 h. Fixed brains were cryoprotected in 30% sucrose in 0.1 M phosphate buffer (PB) for 48 h and frozen while floating on liquid nitrogen. Thereafter, the brains were stored at −70 °C until cryosectioning.

### 2.5. Behavioral Tests

Multiple behavioral tests were used for assessing the general behavior of mice as well as learning and memory functions as presented in Table 3.

**Home cage monitoring**. Home cage monitoring was used to assess possible deviations from normal mouse behavior. First, food pellets, drinking bottles, and enrichment material were removed and the single-housed mouse was let to settle for 10 min. Then, the spontaneous behavior was video-recorded from the side for 10 min. Bouts of grooming and climbing on the grid lid were counted offline.

**Exploratory activity**. In the main set of experiments, the TruScan^®^ (Coulbourn Instruments, Allentown, PA, USA) automated activity monitoring based on infrared photobeam detection was used to assess spontaneous locomotion and exploratory activity. In brief, the system comprises a white-walled observation cage (26 cm × 26 cm × 39 cm) and two rings of photobeam detectors to monitor horizontal and upright positions. A free 10-min exploration was recorded. The analyzed parameters included: ambulatory distance (gross horizontal locomotion), stereotypy time (time engaged in movements that repeatedly break adjacent beams three times), time in the cage center, and vertical time (rearing). In Batch II, a mouse was placed inside an observation arena made of semi-transparent plastic (37 cm × 21 cm with a 15 cm wall) placed inside a 42 cm × 28 cm wood frame with a 16 cm wall. 18 cm). A white paper was placed under the bottom of the plastic arena to improve the contrast. Mouse horizontal movements were tracked with the EthoVision setup (Noldus, Wageningen, the Netherlands) paired with GenICam (Basler, Ahrensburg, Germany) under infrared lighting. A free 60-min exploration was recorded. Eight mice were analyzed at the same time. 

**Nest building**. The nest building test was used to evaluate motor deficits, as well as general changes in the health or welfare of the mice. All enrichment material was removed and replaced by a sheet of tissue paper. The mouse was freely left to build a nest overnight. The outcome was photographed and scored by blinded raters according to the original description by [15] on a scale of 0–4. 

**Rotarod**. The Rotarod test was performed to assess motor coordination and balance deficits. Mice were tested in an automated Rotarod device (Model: LE-8200, Five Station Accelerating Rotarod, Bioseb, France). At the age of 8 months, mice were left to adapt to staying on the rod at the lowest speed (4–5 rpm) for 2 min. At later ages, the test started right away with 4–5 rpm followed by increasing speed in 30 s steps until 40 rpm. The cut-off was set to 6 min. If the mouse fell or rotated with the rod for three rounds without any correcting steps, the test was ended and the time was recorded. 

**Social approach**. A social approach test was performed to analyze possible deviations in the social behavior of mice. The test was performed as previously described by [16] with minor modifications. The test was carried out in the mouse’s home cage (40 cm × 24 cm × 15 cm) with bedding and two cylindrical cages (height 9 cm, diameter 10 cm, half-filled 60 mL glass beaker on the top to add weight) on the opposite ends of the cage. The cages were separated by an opaque plastic wall with a mouse-sized opening. After a 10-min habituation period, a stranger mouse (an ISO mouse of the same sex as the study mouse) was placed in one of two cages. The actual test took 10 min and was video-recorded. The number of nose contacts with the cages and the time sniffing were calculated offline from the recording by a rater blinded to the transplantation group.

**Elevated plus maze**. The elevated plus maze test is a widely used measure of anxiety. The maze comprised four arms (25 cm × 5 cm in Batch I, 40 cm × 5 cm in Batch II) radiating from a central platform (5 cm × 5 cm) and was placed 50 cm (53 cm in Batch II) above the floor. Two of the arms were open, two were closed with a 16 cm (19 cm in Batch II) high wall. The maze was made of black plastic, but the arms were covered with white paper (plastic mat) to give a contrast to the video image. In Batch II, grey plastic was used. The mouse was placed on the central platform, the nose pointed to a closed arm, and video-recorded for 5 min. The number of transitions between the arms and the time spent in the open and closed arms was calculated. The percentage of the total time spent in open arms was analyzed.

**Forced swimming**. This test is widely used to assess behavioral despair. Two 3.0 L glass containers (in the Batch II experiment, 5.0 L) were placed on a table with a wall to block the view between them. The containers were filled with 2.0 L of water (T 22 °C; in the Batch II experiment, 3.0 L). Two mice were tested at the same time. The mice were gently placed on the water surface for 6 min and video-recorded. Immobility (floating) time during the last 4 min was calculated offline by a blinded rater. After the test, the mice were dried with a towel and placed under an infrared lamp or on a heating pad to warm up.

**Spontaneous alternation in a Y-maze**. The test was used for investigating exploratory behavior and cognitive function (related to spatial learning and memory. The maze consisted of three 10 cm x 10 cm compartments that were connected with 5 cm × 5 cm openings as in the letter Y. The maze had 10 cm walls and was made of black plastic. The mouse was placed into the start arm (A) and left to make 20 arm visits undisturbed. The number of alternations as a proxy for working memory (for example, BAC, BCA) was counted. 

**Novel object recognition**. The test was used to evaluate cognition, particularly recognition memory. At the age of 8 months, half of the mice were given object A (50 mL test tube with a yellow paper strip inside) and the other half were given object B (the piston of a standard plastic 20 mL syringe) to familiarize themselves with their home cage (with all enrichment removed) for 20 min. The next day, the mice were given 10 min to explore the same object. Touching the object with the nose or the nose pointing to the object closer to 1 cm was counted as object exploration. Mice that did not spend 20 s in object exploration were excluded from the test. The test phase started 60 min later. The familiar object (A or B) was paired with the new object (B or A), with the position balanced within groups of animals. Exploration of the new vs. familiar objects was video-recorded for 5 min. At 12 months of age, the test was repeated with two new objects, a wooden door knob C and a green square Duplo block D. Novelty preference (%) was calculated as follows:(time at the new object—time at the familiar object) * 100/total exploration time.

**Morris swim task** (**water maze, MWM**). The Morris swim task was used to measure spatial learning and memory. The apparatus was a white plastic wading pool with a diameter of 120 cm filled with water and located in a room containing distal visual cues on the walls. A transparent glass platform (10 × 10 cm) was placed 1.0 cm below the water surface. The temperature of the water was maintained at 20 ± 0.5 °C throughout the experiment, and a short recovery period in a warmed cage was allowed between trials. First, the mice were pretrained for two days to find and climb onto the submerged platform placed between SW and NW quadrants by guiding them through an alley with high walls. During the spatial learning phase (days 1–4), the animals were given five trials/day with 10 min rest between the trials. On day 5, the platform was removed from the pool for the first and the last trials to determine the search bias. The platform location was kept constant, but the starting position varied between four constant locations at the pool rim (N, W, S, E), with all mice starting from the same position in any single trial. If the mouse failed to find the escape platform within 60 s, it was placed on the platform for 10 s by the experimenter. The task was video-recorded with a ceiling camera. The escape latency and swimming speed were calculated for each trial and then averaged across all 5 days of task acquisition. The search bias during the probe trial was measured by calculating the time the mice spent within 30 cm from the center of the former platform position. 

**Odor discrimination**. The test was used for assessing olfactory discrimination learning related to the piriform cortex. Day 1: The mice to-be-tested and separate odor donor mice were kept without a cage change for 5 days to impregnate their odor to the bedding. Every test mouse was given one wooden ball (diameter 2.5 cm) in its home cage and odor donor mice received 2–5 balls according to how many were needed to test all the mice. Balls were left in the cages overnight (from 2 p.m. to 9 a.m.). Day 2: In the morning, the balls were collected into plastic bags individually, to avoid contamination/evaporation of odors, and the bedding in the cages was changed. The cage enrichments were removed at this point. Testing: The mouse cage was placed under the video camera, and two wooden balls were placed near the corners of one end. The left ball was from the cage of a donor mouse and the right ball originated from the mouse’s own cage. The mouse was given 120 s to freely explore the odor-impregnated balls under overhead video recording. Sniffing was counted when the nose of the mouse was >1 cm from the wooden ball. In the evening, again one wooden ball was placed into each home cage overnight. Equal numbers of balls were placed overnight into a plastic bag with cardamom spice. Day 3: Testing was conducted as on Day 2, but this time the left ball was cardamom-odorized and the right one had the mouse’s own odor. Odor preference (%) was calculated as in the novel object recognition task.

**Passive avoidance**. Fear conditioning was tested using the passive avoidance test. The test was performed in a light-dark box (42 cm × 32 cm × 22 cm, Med Associates Inc., Fairfax, VA, USA). One half of the cage was brightly illuminated while the other side with access through a mouse hole was protected from light. On the first day, the mouse was allowed to explore the test arena freely for 5 min. When the mouse entered the dark compartment, a sliding door on the dividing wall with a mouse hole was closed, and the mouse was confined in the dark compartment for 30 s. The mouse received two mild electric shocks (0.30 mA, 2 s) and thereafter was returned to its home cage. On test day 3 (48 h later), the mouse was placed again on the lit side of the two-compartment box. After 30 s, the sliding door was opened and the delay for the mouse to enter the dark compartment (all four paws inside) was again measured. Then the mouse was returned to its home cage or when the total test time of 3 min had elapsed. 

### 2.6. Immunohistochemistry

Twelve series of 30 μm thick coronal sections were cut using an SM2010 R microtome (Leica Biosystems, Nanterre, France) over the entire forebrain and stored in cryoprotectant (30% glycerol, 30% ethylene glycol in Tris-HCl 50 mM, pH 7.4) at −20 °C. One series was used for each combination of antibody stainings. The number of human cells in the cingulate cortex was quantified in sections ranging from +1.1 to −0.2 mm relative to the Bregma. For the quantification of human cells in CA1, the range was −1.3 to −2.5 mm relative to the Bregma. Free-floating sections were washed for 15 min in phosphate-buffered saline (PBS), then in PBS + 0.25% Triton-X100 (PBST). Endogenous peroxidases were quenched with 3% H_2_O_2_ in 10% methanol/PBST, and the sections were washed again 3 times for 10 min in PBST. For BrdU unmasking, sections were incubated in HCl 2N for 50 min and then in borate buffer for 2 × 15 min. For Aβ unmasking, the sections were incubated for 1 h in 100 mM sodium citrate buffer, pH 6.0. After 2 × 15 min washes in PBST, the sections were permeabilized for 30 min in PBS + 0.4 % Triton-X100. After three 5-min rinses in PBST, the first blocking step was performed for 1 h using Mouse on Mouse reagent (Vector Laboratories, Burlingame, CA, USA). Sections were then washed again before being blocked in 10% normal donkey serum or goat serum + 0.1% BSA in PBST (blocking buffer) for 1 h and incubated with primary antibodies (Table 4) overnight at room temperature (RT) in the blocking buffer. Sections were rinsed three times for 10 min in PBST, then incubated for 1.5–2 h with Alexa Fluor-conjugated or biotinylated secondary antibodies in PBST + 10% normal donkey serum or goat serum. The incubation with biotinylated antibodies was followed by 1 h in avidin-biotin-peroxidase complex (1:500; Vector Labs Elite Kit) in PBST. Peroxidase activity was detected using 3,3’-diaminobenzindine-4 substrate (Vector Laboratories, DAB Kit) or nickel-enhanced 3,3’-diaminobenzindine-4 substrate (Vector Laboratories, DAB Kit). The reaction was stopped by rinses in PBST-azide and sections mounted onto subbed slides, dehydrated through alcohols, and coverslipped. In the case of Alexa Fluor-conjugated secondary antibodies, the sections were stained with nuclei marker Hoechst (1:10000) during the first post-staining wash. After two more washes for 10 min in PBST, the sections were mounted on slides in Mowiol solution and coverslipped. For cell counting, 6 to 8 images per mouse were acquired using the Mercator software (Explora Nova, La Rochelle, France) at 20× magnification with a Leica DM6000 B fluorescence microscope (Leica Biosystems). Counting was performed automatically on thresholded images using the analyze particle function of the ImageJ software (RRID:SCR_002285). For colocalization analysis, at least 130 HuNu+ cells per mouse were manually evaluated for co-labeling with BrdU or glial cell markers.

In the Batch II, immunohistochemistry was carried out as follows: 20 μm thick sagittal mouse brain sections preserved in cryoprotectant were washed three times for 10 min in 0.1 M PB, pH 7.4, and overnight at +4 °C with rocking. The next day, the sections were washed again three times for 10 min in PB and mounted on glass slides. The slides were air-dried and stored at −80 °C until staining. At the beginning of the staining, the slides were rehydrated for 5 min in PBS. The staining was carried out in a similar way to Batch 1 except for a few modifications. For washings and reagent dilutions, PBS + 0.05% Tween-20 was used. For blocking, 5% normal goat serum in PBS + 0.05% Tween-20 was used. Alexa Fluor-conjugated secondary antibodies were diluted 1:800. Sections were coverslipped using Fluoromount mounting medium (Thermo Fisher Scientific).

### 2.7. Reverse Transcription Quantitative Real-Time PCR (RT-qPCR) and RNA Sequencing

Hippocampal RNA (*PSEN1* ΔE9 *n* = 5, CTRL *n* = 7) was extracted using TRI Reagent (Merck) according to the manufacturer’s instructions and thereafter treated with DNAse I (molecular biology grade, Thermo Fisher Scientific) for 30 min at 37 °C. RNA was further purified using RNeasy Mini Kit (Qiagen, Hilden, Germany) according to the manufacturer’s instructions. RNA integrity was determined using TapeStation 4200 (Agilent). The RIN values of all RNA samples were above 8. In the Batch II experiment, hippocampal RNA was extracted using the PARIS kit (Thermo Fisher Scientific).

cDNA was synthesized using Maxima reverse transcriptase (Thermo Fisher Scientific), and relative mRNA expression was determined using validated Taqman primers (Table 5) and Maxima Probe qPCR Master Mix (Thermo Fisher Scientific) on the Bio-Rad CFX96 Real-Time System (Bio-Rad, Hercules, CA, USA). The results were normalized to mouse *Gapdh* or human *GAPDH* expression using the Q-gene program (Equation (2)) [17]. RNA-sequencing libraries were produced with NEBNext Ultra II Directional RNA Library Prep Kit (New England Biolabs, Ipswich, MA, USA) (*PSEN1* ΔE9 *n* = 5, CTRL *n* = 6). Libraries were sequenced with NextSeq500 (Illumina, 1 × 75bp, high output) using the services of the Biomedicum Functional Genomics Unit at the Helsinki Institute of Life Science and Biocenter Finland, University of Helsinki. Sequence reads were aligned simultaneously to the human genome (GRCh.38) and mouse genome (GRCm.38) using STAR aligner v2.7.8a [18] and annotated to gene exons with HTSeq v0.11.2 [19] (RRID:SCR_005514). Nucleotide variation between the species allowed us to distinguish between sequence reads of the mouse or human origin. On average, 78.34% of total reads had a unique best alignment on either genome or were usable for HTSeq annotation. Differential gene expression (DEG) analysis of *PSEN1* ΔE9 cell-transplanted versus CTRL-transplanted hippocampi was performed using the R package DESeq2 v1.32.0 method [20] (RRID:SCR_015687). DEG data that met the *p*-value < 0.05 and fold change ≥ 1.2 (in both directions) cutoff were further analyzed using QIAGEN Ingenuity Pathway Analysis (IPA) [21]. IPA identified the nervous system-specific pathways from the QIAGEN IPA library that were most significant to the dataset by first calculating a ratio of the number of molecules from the dataset that map to a pathway divided by the total number of molecules that map to the same pathway. Second, a right-tailed Fisher’s Exact Test was used to calculate a *p*-value determining the probability that the association between the genes in the dataset and the pathway is explained by chance alone. Z-score ≤ −1.99 or ≥ 1.99 was used as a cutoff for a directional change. Cell type enrichment analysis was conducted by comparing DEGs from our dataset (*p* < 0.05) to the CellMarker database [22] augmented with co-expression RNA-seq data from ARCHS4 [23] using Enrichr online tool [24] (RRID:SCR_001575). The heatmap of oligodendrocyte-associated genes was created using the Heatmapper online tool (RRID:SCR_016974; www.heatmapper.ca; accessed on 12 December 2022) [25].

### 2.8. Aβ Measurement by ELISA

Freshly frozen cortices (*PSEN1* ΔE9 male *n* = 6, female *n* =3, CTRL male *n* = 5, female *n* = 6) were homogenized in 10× volumes of TBS supplemented with Pierce protease inhibitor (Thermo Fisher Scientific) using a motorized plastic pestle and kept for 30 min on a rotator at 4 °C. The samples were centrifuged for 30 min at 16,000 × *g* at 4 °C, and supernatants were collected to assess the level of soluble Aβ species. Insoluble Aβ was extracted from the remaining pellet by incubating in 8× volumes of 5 M guanidine-HCl/50 mM Tris · HCl, pH 8.0 for 2 h at RT on a shaker at 300 rpm. The levels of Aβ42 and Aβ40 were measured by Human Aβ42 and Human Aβ40 ELISA kits, respectively (R&D Systems, Minneapolis, MN, USA), and normalized to dry tissue weight. 

### 2.9. Western Blotting

Freshly-frozen hippocampi from female mice (*PSEN1* ΔE9 *n* = 4, CTRL *n* = 5) were homogenized in 10× volume of 1× Laemmli sample buffer supplemented with Pierce protease and phosphatase inhibitors (Thermo Fisher Scientific) using a motorized plastic pestle. The samples were denatured for 5 min at 95 °C, and proteins were resolved by gel electrophoresis using 4–20% precast TGX Mini-Protean gels (Bio-Rad). Western blotting was performed using the Trans-Blot Turbo Transfer system (Bio-Rad). The membranes were blocked in 5% non-fat milk in TBS + 0.05% Tween-20 (TBST) and probed overnight with primary antibodies (Table 4) diluted in 5% BSA/TBST + 0.02% sodium azide. The membranes were washed three times for 10 min in TBST and reprobed for 1 h with secondary HRP-conjugated antibodies (Table 4) diluted in 5% non-fat milk in TBST. The membranes were then washed 3 × 10 min in TBST followed by 5 min in TBS. For band detection, Pierce ECL Plus Western blotting substrate (Thermo Fisher Scientific) was added to the membranes, and the signal was visualized using G:BOX Gel and Plot Imaging System (Syngene, Bangalore, India).

### 2.10. Statistics

Statistical analyses were performed with GraphPad Prism 9.2.0 (Insight Partners, New York, NY, USA) or IBM SPSS Statistics software, version 27.0 (IBM Corp., Armonk, NY, USA) using Student’s t-test, repeated measures ANOVA, or two-way ANOVA with Šidák’s-corrected posthoc tests. Statistical significance was assumed at *p* < 0.05. All data are shown as mean ± SEM. Significant outliers were detected and excluded from the analysis using the GraphPad outlier test.

### 2.11. Data Availability

Appendix A show the full list of human (Appendix A) and mouse (Appendix A) differentially expressed genes with raw expression values. Appendix A show the list of significantly enriched pathways in human (Appendix A) and mouse (Appendix A) datasets identified by IPA software. The raw RNA-sequencing data are uploaded in full to the Gene Expression Omnibus database: GSE221027. Appendix A show the data from all the behavioral tests conducted that were not included in the main part of the manuscript. Appendix A shows the behavioral data presented in Figure 2 as stratified by the parental iPSC line of the transplanted cells. Appendix A shows representative images of Aβ-positive aggregates in transplanted mouse brains and the quantification data. Appendix A shows raw Western blot images.

## 3. Results

### 3.1. PSEN1 ΔE9 Mutation in Human Glial Progenitors Improves Spatial Learning and Memory at 16 Months after the Transplantation

To evaluate the effect of the *PSEN1* ΔE9 mutation in transplanted human glial progenitor cells on mouse locomotion, emotional state, and cognitive function, we designed a wide panel of behavioral tests (Table 3). It is known that the transplanted glial progenitors proliferate and gradually replace mouse glia in the cortex over the course of at least 12 months [11]. Additionally, it normally takes at least 12 months to see cognitive deficits in AD transgenic mouse models [26]. Therefore, we chose the ages of 8, 12, and 16 months to evaluate behavioral changes. At the age of 8 and 12 months, the mice were monitored in their home cage for grooming and climbing and underwent the following behavioral tests: explorative activity (TruScan), nest building, Rotarod, social approach, elevated plus maze, forced swimming, spontaneous alternation in the Y-maze, and novel object recognition. Spontaneous exploratory activity usually decreases during repeated visits to the same environment. Therefore, the mice were tested in the TruScan box on two different days at the age of 8 months to make the test environment familiar to the 8-month-old mice as it would be familiar to the mice at later ages. At the age of 16 months, in addition to the above-mentioned tests, mice underwent Morris swim task (watermaze), odor discrimination, and passive avoidance tests. As there are major sex differences in the behavioral test outcomes in general, we ran the statistical testing separately for males and females.

In 8-month-old mice of both genotypes, the ambulatory distance in TruScan drastically decreased (males: F_1,18_ = 151.8, *p* < 0.001; females: F_1,18_ = 149.4, *p* < 0.001; Figure 2A,D) and stereotypy time increased (males: F_1,18_ = 68.8, *p* < 0.001; females: F_1,18_ = 53.7, *p* < 0.001; Figure 2B,E) when the mice were exposed to the same environment on a second day. On the other hand, during re-exposure to the same environment at 12 and 16 months of age, the habituation effect was attenuated, speaking for forgetting with advancing age. The ambulatory distance grew from the second visit at 8 months to the fourth visit at 16 months in both iPSC genotypes (males: F_2,16_ = 11.7, *p* = 0.001; females: F_2,16_ = 8.5, *p* = 0.003; Figure 2A,D), while the stereotypy time decreased correspondingly (males: F_2,16_ = 7.7, *p* = 0.004; females: F_2,16_ = 14.7, *p* < 0.001; Figure 2B,E). During the last visit at the age of 16 months, male, but not female, *PSEN1* ΔE9-transplanted mice displayed a significantly increased stereotypy behavior as compared to the CTRL mice (Figure 2B, Šidak’s posthoc test, *p* = 0.024). This effect was not seen in younger mice. There was no significant genotype effect on ambulatory distance in TruScan at any age tested.

In the forced swim test, *PSEN1* ΔE9 –transplanted male mice exhibited an overall longer immobility time than the controls (F_1,17_ = 9.0, *p* = 0.008). Šidak’s posthoc tests showed that the difference reached statistical significance at the age of 16 months (Figure 2C, *p* = 0.01). Female mice did not show a significant age (*p* = 0.38) or iPSC genotype effect (*p* = 0.65) in the forced swim test (Figure 2F). Interestingly, irrespective of the mouse sex, the immobility time correlated strongly and positively with the stereotypy time (Spearman’s rho = 0.45, *p* = 0.004, *n* = 39), suggesting that both measures reflect a common behavioral feature. 

The novel object recognition (NOR) test of short-term object memory (60 min) showed that the preference for the novel object strongly declined at the age of 16 months in both male and female mice (*p* < 0.001) without a significant genotype effect (*p* > 0.39, Appendix A). In the Morris swim task of spatial learning performed at the age of 16 months, male *PSEN1* ΔE9 glia-transplanted mice did not differ from the CTRL mice in the task acquisition over all 5 days (F_1,17_ = 3.3, *p* = 0.09). However, when considering only the last two days, they performed better (F_1,17_ = 5.5, *p* = 0.03) than the CTRL mice (Figure 2G). The female *PSEN1* ΔE9 transplanted mice did not differ from CTRL mice in their escape latency either across all 5 days of task acquisition (Figure 2H; F_1,16_ = 1.4, *p* = 0.24) or during the last two days (F_1,16_ = 2.2, *p* = 0.16). Importantly, the swim speed did not significantly differ between the iPSC genotypes in either sex (Appendix A, *p* > 0.24). Both male (Figure 2I; t_17_ = 2.7, *p* = 0.02) and female (t_16_ = 2.5, *p* = 0.03) *PSEN1* ΔE9 glia-transplanted mice displayed a stronger search bias to the former platform position in the second probe trial than the CTRL mice, indicative of better spatial memory. In the passive avoidance test, another spatial memory test performed at the age of 16 months, male, but not female, mice exhibited significantly increased latency to enter the dark compartment 48 h after the electric shock (F_1,16_ = 7.9, *p* = 0.012; Appendix A), suggesting a better long-term memory. There was a trend towards a longer latency in *PSEN1* ΔE9 glia-transplanted male mice as compared to the CTRL (trial x genotype F1,16 = 2.42, *p* = 0.14), again suggesting some improvement in spatial memory. 

No significant genotype effect was seen in the Rotarod, home cage climbing and grooming, elevated plus maze, nest building, social approach, odor discrimination, or spontaneous alternation in Y-maze (Appendix A). When we further stratified the results of the stereotypy behavior in TruScan, forced swim test, and Morris swim task by the parental iPSC line of the transplanted cells (AD2 vs. AD3), we observed no clear difference between the two *PSEN1* ΔE9—CTRL pairs (Appendix A).

### 3.2. Transplanted Human Glial Cells Were Distributed throughout the Mouse Brain and Expressed Predominantly OPC Markers

To evaluate the engraftment and distribution of the transplanted human glial progenitor cells, we sacrificed the transplanted animals at 18 months of age, and immunostained brain sections for human nuclear (HuNu) antigen. HuNu-positive cells were widely distributed throughout the brain (Figure 3A). The lowest engraftment was observed in the thalamus, hypothalamus, posteromedial and lateral cortical amygdaloid nuclei, and piriform cortex. All mice exhibited a homogenous HuNu staining, except one mouse transplanted with AD cells and three mice transplanted with CTRL cells (excluded from all the analyses). Two brain areas were then chosen for the detailed analysis of human cell engraftment, the cingulate cortex (Cg) and the CA1 area of the hippocampus. The cingulate cortex is involved in sustaining attention and motivational control of cognitive tasks [27,28], while the CA1 area plays a key role in spatial memory formation. The quantification of the density of HuNu-positive cells in the cingulate cortex (Cg) and the CA1 area of the hippocampus (Figure 3B) suggested a similar integration of CTRL-grafted cells in male and female mice, with 489 ± 52 cells per mm^2^ in the Cg and 403 ± 59 cells per mm^2^ in the CA1 area. Interestingly, the density of *PSEN1* ΔE9 mutant HuNu-labeled human cells was significantly lower than that of CTRL cells in male mice (F_1,14_ = 7.844, *p* = 0.014). No significant *PSEN1* genotype effect on human cell density was observed in female mice (F_1,15_ = 1.222, *p* = 0.29).

To determine the phenotype of the transplanted human cells, brain sections were co-stained with HuNu antibody and the antibodies for OLIG2 (a marker for OPCs and oligodendrocytes), PDGFRα (a marker for OPCs), S100β (a marker for astrocytes and OPCs), and GFAP (a marker for astrocytes). Qualitatively, there were numerous cells co-expressing HuNu antigen and OLIG2, PDGFRα, or S100β (Figure 3C). However, there were only a few cells clearly co-expressing HuNu antigen and GFAP (Figure 3C), mostly in the corpus callosum area (Figure 3D). The quantification of HuNu and PDGFRα co-staining in the CA1 area of the hippocampus showed that about 80% of HuNu-positive cells also expressed PDGFRα, suggesting that these cells were OPCs (Figure 3F). There was no difference between genotypes. 

To check if human cells were still proliferating in vivo when the grafted animals reached 18 months of age, brain sections were co-stained with HuNu and BrdU-specific antibodies (Figure 3E). The quantification analysis showed that 19 ± 6 % of CTRL HuNu-positive cells in the Cg and 30 ± 6 % in the CA1 area (Figure 3G,H) were also BrdU-positive, suggesting that some grafted cells were still proliferating. This finding is consistent with the OPC identity of the grafted cells. Interestingly, this proliferation was significantly reduced in *PSEN1* ΔE9 mutant cells as compared to CTRL cells transplanted into male mice (F_1,13_ = 4.92, *p* = 0.045). No significant *PSEN1* genotype effect on the proliferation of human cells was observed in female mice (F_1,16_ = 0.038, *p* = 0.85). Overall, these results indicate that the transplanted cells integrated into the mouse brains and retained proliferative nature even at 18 months of age. 

### 3.3. PSEN1 ΔE9 Mutant Human Cells Converted a Higher Proportion of Aβ42 into Insoluble Form Than the Control Cells

To evaluate whether grafted human glial cells produced Aβ deposits, we immunostained brain sections with anti-Aβ peptide antibody (Appendix A). Only occasional Aβ deposits (arrows) were seen in both CTRL and *PSEN1* ΔE9 glia-transplanted brains in the vicinity of HuNu-positive cells (arrowheads). There was no significant difference in the absolute number of deposits per brain hemisphere between CTRL and *PSEN1* ΔE9 glia-transplanted mice (Appendix A). 

To investigate whether there was a *PSEN1* genotype effect on the content of soluble and insoluble Aβ species in the brain, we sequentially homogenized frozen cortices, first in TBS to extract soluble Aβ species (monomers and small oligomers) followed by 5 M guanidine-HCl to extract large insoluble Aβ aggregates. The content of Aβ42 and Aβ40 in cortical extracts was measured by commercial ELISA kits. In the TBS-soluble fraction, low, but detectable, levels of both Aβ42 and Aβ40 (Figure 4A,D) were observed. Interestingly, the *PSEN1* ΔE9 mutation significantly decreased cortical levels of soluble Aβ42 (F_1,14_ = 10.35, *p* = 0.006) but not Aβ40 (*p* = 0.39) and did so more in male (Šidak’s posthoc *p* = 0.02) than female mice (*p* = 0.82). Consistently with a higher propensity for aggregation of Aβ42, only Aβ42 was detected in the guanidine fraction, without a significant sex or genotype effect (Figure 4B). Also, total levels of cortical Aβ42 (TBS and guanidine fractions combined) were not affected by mouse sex or iPSC genotype (data not shown). However, when calculating the ratio of Aβ42 content in the guanidine fraction in relation to the content in the TBS fraction, we found that the ratio was significantly increased by both female sex (F_1,16_ = 8.571, *p* = 0.001) and *PSEN1* ΔE9 genotype (F_1,16_ = 6.983, *p* = 0.018, Figure 4C). These results suggest that 1) the *PSEN1* ΔE9 mutation did not significantly affect Aβ42 and Aβ40 production in human glia; 2) *PSEN1* ΔE9 mutant glia converted a higher proportion of Aβ42 to an insoluble form, resulting in significantly lower levels of soluble Aβ42 in *PSEN1* ΔE9-transplanted males. Cortical levels of soluble Aβ42 inversely correlated with time spent in the target zone in the second probe test (Spearman’s Rho = −0.567, *p* = 0.014, *n* = 18, Figure 4E) and directly correlated with escape latency in the Morris swim task on day 5 (Spearman’s Rho = 0.545, *p* = 0.019, Figure 4F). This finding suggests that the soluble Aβ42, and not the insoluble fraction, may have contributed to spatial memory impairment in the Morris swim task. Furthermore, the levels of soluble Aβ42 inversely correlated with immobility time in the forced swim test in male mice (Spearman’s Rho = −0.661, *p* = 0.044, *n* = 10, Figure 4G), suggesting that the mouse motivation to find an escape in the Morris swim task may have been affected by soluble Aβ42 species. 

### 3.4. PSEN1 ΔE9 Mutant Human Glial Cells Exhibit an Altered Transcriptome

To further investigate the mechanisms underlying altered mouse behavior and the aggregation of Aβ42, we RNA-sequenced the hippocampi of six CTRL-transplanted and five *PSEN1* ΔE9-transplanted male mice. *PSEN1 ΔE9* mutant human cells exhibited a significantly altered transcriptome with 462 upregulated human genes and 381 downregulated human genes (threshold: 1.2-fold change, *p* < 0.05; Figure 5A; Appendix A). Ingenuity pathway analysis (IPA) showed that the only significantly activated canonical pathway (Z-score ≥ 1.99) in *PSEN1 ΔE9* mutant cells was the neurovascular coupling signaling pathway (Z = 2.324, *p* = 0.032; Figure 5B; Appendix A). Top inhibited canonical pathways were the oxytocin in brain signaling pathway (Z = −2.668, *p* = 0.002; Figure 5C; Appendix A), estrogen receptor signaling (Z = −2.982, *p* = 0.029), oxytocin signaling (Z = −2.236, *p* = 0.011), and G beta gamma signaling (Z = −2.111, *p* = 0.010). IPA analysis of upstream gene regulation suggested the activation of the transcriptional regulator TCF7L2 (TCF4, Z = 4.243, *p* < 0.001), an important effector of the Wnt/β-catenin signaling pathway. No significant difference in the level of *TCF7L2* mRNA expression was observed between the groups (Figure 5D).

IPA upstream analysis also suggested inhibition of transcriptional regulator HTT (huntingtin; Z = −2.381, *p* < 0.001) primarily associated with the Huntington’s disease. There was no significant difference in the levels of *HTT* mRNA either (data not shown). NOTCH1 expression was significantly downregulated by the *PSEN1* ΔE9 mutation in the transcriptomics dataset (adj *p* = 0.048; Figure 5D). Interestingly, the *PSEN1* ΔE9 human cells exhibited a higher expression of myelinating oligodendrocyte signature genes [29,30,31], although the effect was stronger in cells originating from the AD2 parental line in comparison to AD3 (Figure 5D). The sex of the parental line (male AD2 vs. female AD3) may have contributed to the observed variation. Further, we validated significant upregulation of several genes by RT-qPCR in *PSEN1* ΔE9 human glia, including *PLP1, CNTN2, TF, SEPP1, APP, MEG3, TRIM4*, and *TCEAL5* (Figure 5E). Out of these genes, four (*PLP1* (proteolipid protein 1), *CNTN2* (contactin 2), *TF* (transferrin), and *SEPP1* (selenoprotein P)) are enriched in oligodendrocytes in the human brain and are involved in myelinogenesis. Upregulation of *MEG3* [32], *TRIM4* [33], and *TCEAL5* [34] were reported in different models of Huntington’s disease with an intracellular aggregation of mutant HTT, suggesting a link between *PSEN1* ΔE9 and mutant HTT-induced transcriptomic changes. 

Engrafted *PSEN1* ΔE9 mutant glial cells also induced some changes in mouse hippocampal transcriptome as compared to the CTRL cells. There were 140 significantly upregulated and 67 significantly downregulated genes (threshold: 1.2-fold change, *p* < 0.05; Figure 6A; Appendix A). Out of 67 downregulated genes, 30 have unknown function, 6 are poorly characterized lnc RNAs, and 7 exhibited a very low expression level. The relative expression of the rest 24 downregulated genes and of vasculature-associated upregulated genes is shown in Figure 6A. Top activated canonical pathways in *PSEN1* ΔE9 glia-transplanted hippocampi were wound healing signaling pathway (Z = 2.646 (biased), *p* < 0.001; Figure 6B; Appendix A), acute phase response signaling (Z = 2.000, *p* = 0.008), and production of nitric oxide and reactive oxygen species in macrophages (Z = 2.000, *p* = 0.011). Among the pathways related to diseases and functions, survival of neural cells (Z = 1.991, *p* = 0.013; Figure 6C; Appendix A) and cell viability (Z = 1.991, *p* = 0.014) was significantly increased. Cell type enrichment analysis using the Enrichr online tool revealed that the upregulated genes were significantly enriched in smooth muscle cells (odds ratio 13.37, adj *p* < 0.001; Appendix A) and brain pericytes (odds ratio 11.61, adj *p* < 0.001), but not in brain endothelial cells (adj *p* = 0.247) or macrophages/microglia (adj *p* = 0.542), suggesting that the vascular mural cells were the mouse cell type most affected by the *PSEN1* ΔE9 mutation in grafted human glia. 

Remarkably one of the highest-expressing genes in engrafted human cells was *VCAN*. *VCAN* encodes for a large chondroitin sulfate proteoglycan (CSPG) versican shown to be involved in both memory retrieval [35] as well as the proliferation and migration of smooth muscle cells/vascular remodeling [36,37] and considered to be a marker for human OPCs [38]. Consistently with the OPC identity of human cells, *VCAN* was the only extracellular matrix gene expressed at a higher level in human cells as compared to mouse cells in the hippocampus when both genes were normalized to the mouse housekeeping gene *Gapdh* (Figure 6D; Appendix A). We further calculated the ratio of human to mouse *VCAN* gene expression and found that the ratio was significantly higher in *PSEN1 ΔE9*-transplanted mice (*p* = 0.023; Figure 6E). We further assessed the expression of other mouse CSPG-related genes by RT-qPCR and found a significant downregulation of mouse *Ncan* and *Fibcd1* genes in *PSEN1 ΔE9*-transplanted hippocampi as compared to the isogenic CTRL (Figure 6D). The *Ncan* gene encodes for a CSPG neurocan, the main component of perineuronal nets stabilizing synapses. A very recent study has identified the fibrinogen C domain containing 1 (FIBCD1) as an endocytic receptor for CSPGs predominantly expressed in the CA1 area of the hippocampus within the brain [39]. Thus, the *PSEN1* ΔE9 mutation in transplanted human glia dysregulated the expression of CSPG signaling molecules in the mouse hippocampus. 

The Western blot analysis of the hippocampal lysates obtained from female mice showed that the presence of *PSEN1 ΔE9* glia increased the phosphorylation of ERK (p44/42) at Thr202/Tyr204 (Figure 6F,J) indicating activation of this kinase. The within-group variation could not be explained by the origin of the parental iPSC line. ERK signaling pathway regulates cell proliferation, growth, and survival, and thus its activation is consistent with the activation of acute phase response and cell viability pathways in *PSEN1 ΔE9*-transplanted hippocampi.

### 3.5. Human Glial Transplantation Decreased Immobility Time in the Forced Swim Test as Compared to the Sham-Transplanted Controls

Since our main set of animals did not include sham-operated mice, we did not know whether the CTRL or *PSEN1 ΔE9*-transplanted group was closer to the behavior of the non-transplanted mice of the same genetic background. To control for this, we performed another set of transplantations where we injected glial progenitors derived from two non-diseased male iPSC lines (unrelated to the lines used in the main set) or PBS only to obtain sham-operated controls (Figure 7A). At the age of 16 months, these mice were subjected to a small panel of behavioral tests, including exploratory activity in the open arena, elevated plus maze, and the forced swim test. To study exploratory activity, mice were allowed to freely explore an open arena for 60 min while the ambulatory distance was recorded in 10-min blocks. As expected, during the first 10 min the mice were actively exploring the arena, but after becoming familiar with it, their locomotive activity started to decrease reaching significance by the second 10-min block in the females (*p* < 0.001; Figure 7C) and by the fourth 10-min block in the males (*p* = 0.003; Figure 7B), indicating normal habituation. Human cell-transplanted females habituated faster than the sham controls (time x group interaction during the first 30 min F_2,52_ = 10.6, *p* < 0.001) while males habituated slower (time x group interaction during the first 30 min F_2,20_= 6.4, *p* = 0.007). In the forced swim test, human glia–transplanted mice exhibited an overall shorter immobility time than sham-operated controls (F_1,35_ = 8.2, *p* = 0.007). Šidak’s posthoc tests showed that the difference reached statistical significance in males (*p* = 0.03), but not females (Figure 7D, *p* = 0.3). As expected, the mean immobility time of the transplanted male mice (63.0 ± 62.0 s) was similar to the time observed in the CTRL male group of the main set of animals (67.1 ± 38.8 s). Interestingly, the immobility time displayed by the sham-operated males (137.7 ± 47.3 s) was remarkably similar to the results observed in the *PSEN1* ΔE9 group (131.4 ± 42.2 s). These data indicate that the transplantation of human glia decreases the immobility of male mice in the forced swim test, which is now believed to be a coping strategy in a stressful setting [40]. On the other hand, the *PSEN1* ΔE9 mutation in the transplanted human glia restored the normal immobility response. No significant differences between the transplanted mice and sham controls were observed in the elevated plus maze (Figure 7E). To determine whether the presence of CTRL human cells increases or decreases mouse *Ncan* and *Fibcd1* expression in comparison to shams, we isolated hippocampal RNA and ran RT-qPCR analysis (Figure 7G,H). There was no significant difference in the expression of *Ncan* and *Fibcd1* between the groups in the males, suggesting that the expression of CSPG-related genes in *PSEN1* ΔE9-mutant males was not similar to shams. Interestingly, in the females transplanted with human cells both these genes were significantly downregulated as compared to shams correlating with faster habituation in the open arena. We verified the presence of human cells in the hippocampus by HuNu staining (Figure 7F). On average, the engraftment levels in this batch of transplantations were lower than in the main batch. Nevertheless, HuNu-positive cells were clearly present in the CA1 area of the hippocampus (Figure 7F, arrow).

## 4. Discussion

We have demonstrated that both CTRL and *PSEN1* ΔE9 mutant human glial progenitor cells can survive in the immunodeficient mouse brain for at least 18 months after the transplantation and generate different types of glial cells predominantly of OPC/oligodendrocyte origin characterized by PDGFRα and OLIG2 immunoreactivity with occasional GFAP-positive astrocytes in the corpus callosum. This finding is consistent with earlier reports by Windrem and coworkers [11,12]. Interestingly, the *PSEN1* ΔE9 mutation significantly decreased human cell proliferation and possibly migration and increased the expression of myelination-associated genes in male mice suggesting that it affected OPC differentiation into oligodendrocytes. We also found that the *PSEN1* ΔE9 mutation in transplanted human cells increased stereotypic behavior in an explorative activity test (TruScan) and immobility time in the forced swim test in aged male, but not female, mice to the level exhibited by sham-operated non-transplanted controls. Most surprisingly, we found that *PSEN1* ΔE9 glia-transplanted aged male mice displayed significantly improved spatial learning in the Morris swim task as compared to the CTRL glia-transplanted males. Moreover, both male and female *PSEN1* ΔE9 glia-transplanted aged mice displayed a stronger search bias to the former platform position than the controls, indicating a better spatial memory.

The *PSEN1* ΔE9 mutation is known to increase the ratio of aggregation-prone Aβ42 to shorter peptides Aβ40 and Aβ38. However, there have been conflicting reports regarding the effect of *PSEN1* ΔE9 on total levels of Aβ42 [4,5,6,7]. Contrary to our previous data obtained in in vitro differentiated iPSC-derived astrocytes [9], we did not find a significant effect of the *PSEN1* ΔE9 mutation on total Aβ42 levels in the cortex but saw a significant decrease in the levels of TBS-soluble Aβ42 in *PSEN1* ΔE9-transplanted male mice. This decrease in the levels of soluble Aβ42 could be explained by both a reduction in the numbers of human cells in the cortex and an increased formation of TBS-insoluble Aβ42 aggregates. Additionally, we cannot exclude the possibility that Aβ degradation/clearance pathways were upregulated in *PSEN1* ΔE9-transplanted mice. Notably, the levels of soluble Aβ42 in females were on average significantly lower and the ratios of insoluble to soluble Aβ42 were significantly higher than in males. This finding is in accordance with previous studies reporting faster amyloid plaque accumulation in the cortex and hippocampus of female AD transgenic mice as compared to males [41,42] and could help to explain why we saw stronger behavioral differences in the male mice. It is important to add that the levels of Aβ42 we detected in our human glia-transplanted mice were much lower in comparison to those normally detected in aged AD transgenic mice. For example, in our *APP*swe *PSEN1* ΔE9 transgenic mouse model, we normally detect on average 5× higher levels of soluble Aβ42 and 10,000× higher levels of insoluble Aβ42. Therefore, it is not surprising that we could detect only occasional very small Aβ deposits by immunohistochemistry and did not see a significant difference between the transplantation groups. 

It has been shown that TBS-soluble Aβ oligomers are more toxic to neurons than insoluble amyloid plaques and are enough to cause memory impairment [43,44,45]. Thus, increased Aβ42 aggregation could be a protective mechanism for decreasing Aβ toxicity. Indeed, cortical levels of TBS-soluble, but not insoluble, Aβ42 directly correlated with the spatial memory impairment in the Morris swim task and inversely correlated with immobility time in the forced swim test in males, which is consistent with an earlier forced swim test study in the 3 × Tg mouse model of AD also using male mice [46]. Torres-Lista & Gimenez-Llort showed that while non-Tg mice displayed a shift towards immobility after the first 2 min of vigorous swimming, 3 × Tg mice did not show a behavioral switch, which resulted in overall lower immobility time. Notably, the behavior of *PSEN1* ΔE9-transplanted males in the forced swim test was very similar to that of sham-operated mice that did not produce any human Aβ42 at all. However, we cannot exclude that other human cell factors could have affected the behavioral results.

How did the *PSEN1* ΔE9 mutation and female sex affect Aβ42 aggregation? The mouse brain is a complex environment with different types of glia and a dense extracellular matrix that can affect the levels and conformation of Aβ species. We detected a very strong upregulation of human long non-coding (lnc) RNA *MEG3* in *PSEN1* ΔE9-transplanted male hippocampi as well as in iPSC-derived astrocytes in vitro [10]. *MEG3* has previously been shown to promote the aggregation of the mutant huntingtin gene in the neuroblastoma cell model of Huntington’s disease [32]. Moreover, a recent preprint reported a strong *MEG3* upregulation in human iPSC-derived neurons that developed sarcosyl-insoluble tau filaments when transplanted into the *APP*-transgenic mouse brain [47]. Thus, the upregulation of *MEG3* could be one potential uncharacterized mechanism behind *PSEN1* ΔE9 mutation-induced Aβ42 aggregation. Moreover, transcriptome analysis showed that the expression of several human myelin-associated genes, including *PLP1*, *CNTN2*, *TF*, and *SEPP1* (all were also confirmed by RT-qPCR) were upregulated in *PSEN1* ΔE9 mutant human cells. Of these genes, the *CNTN2* product directly binds to APP and affects its processing [48], while transferrin encoded by *TF* [49] and selenoprotein P encoded by *SEPP1* [50] can bind to Aβ and modulate its aggregation either directly or by regulating the levels of iron, copper, and zinc in the tissue. It has been suggested that it is the higher levels of extracellular zinc that promote amyloid aggregation in females [51,52], and selenoprotein P has recently been reported to be involved in zinc homeostasis [50,53].

Why did the *PSEN1* ΔE9 mutation increase the expression of myelination-related genes? In addition to affecting APP processing, the *PSEN1* ΔE9 mutation reduces Notch1 signaling [5,7,54] and stabilizes β-catenin [55,56]. Notch1 promotes OPC proliferation but inhibits their maturation into myelinating oligodendrocytes [57,58], while Wnt/β-catenin signaling may have the opposite effect [59]. Our transcriptomics analysis showed a reduction in *NOTCH1* mRNA expression in *PSEN1* ΔE9-grafted human cells and also a very strong upregulation in long non-coding (lnc) RNA *MEG3*, which represses the Notch pathway [60]. On the other hand, IPA analysis of upstream gene regulation suggested the activation of TCF7L2, an important effector of the Wnt/β-catenin signaling pathway. Thus, the effect of the *PSEN1* ΔE9 mutation on human cell proliferation and expression of oligodendrocyte-related genes was likely synergistically mediated by both Notch and Wnt signaling pathways. 

An upregulation of oligodendrocyte-related genes and an increased rate of oligodendrogenesis have also been reported in *APP* and/or *PSEN1*-mutant mice at the early stage of pathology [61,62,63]. Interestingly, a recent human single-nuclei transcriptomics study showed that the proportion of OPCs is decreased, and the proportion of mature oligodendrocytes is increased in the entorhinal cortex of AD patients [64]. The study by Wang and coworkers [65] has reported that the generation of new myelin is dramatically decreased in 18-month-old mice and that the stimulation of myelination can improve spatial memory in aged mice. Thus, the upregulation of oligodendrocyte-related genes in our study could be a mechanism serving to protect memory function in the aged brain. 

We also found that the *PSEN1* ΔE9 mutation was associated with the dysregulation of CSPG signaling in the mouse hippocampus. The expression of mouse *Ncan* and *Fibcd1* genes was significantly downregulated in the *PSEN1* ΔE9-transplanted animals. Neurocan encoded by the *Ncan* gene is the main component of perineuronal nets stabilizing synapses, and it also prevents remyelination in brain injury conditions. The lower level of expression of *Ncan* and its receptor *Fibcd1* could enable a higher level of synaptic plasticity. A genetic variant in the human *NCAN* gene increasing the level of neurocan expression was recently found to be associated with worsened hippocampal memory function in healthy humans [66]. Moreover, FIBCD1 has recently been demonstrated to be involved in hippocampal-dependent learning in mice [39]. Altogether, these gene expression changes are in line with better spatial memory function in *PSEN1* ΔE9-mutant glia-transplanted mice observed in our study. Furthermore, consistently with this notion, we observed lower levels of *Ncan* and *Fibcd1* expression in the hippocampi of human glia-transplanted females that exhibited faster habituation in the open arena in the second batch of the transplantations. 

Importantly, we also found that the *PSEN1* ΔE9 mutation upregulated neurovascular coupling signaling in human cells and increased the expression of smooth muscle cell/pericyte-associated genes in surrounding mouse cells. These findings suggest that *PSEN1* ΔE9-transplanted mice may have exhibited an alteration in neurovascular coupling, the main function of which is to provide adequate oxygen and nutrient supply to firing neurons.

## 5. Conclusions

In sum, we have shown that human glial progenitors can survive and proliferate in the mouse brain for at least 18 months, give rise predominantly to OPCs, produce human Aβ40 and Aβ42, and attenuate the immobility response in the forced swim test in male mice. We have also shown that the *PSEN1 ΔE9* mutation in transplanted human glia had multiple effects decreasing the proliferation of human cells, decreasing the levels of soluble human Aβ42, increasing the expression of myelin-related genes, and inducing wound healing and vasculature development pathways in host mouse cells. These changes were associated with improved spatial learning and memory, especially in male mice. The main advantage of our model is that it allowed the studying of *PSEN1* ΔE9 mutation-induced changes in a specific cell type (glial progenitors) in the physiological conditions of a living brain and to better understand the role of the cells of oligodendrocyte lineage in AD. However, our data also show that experimental transplantation of human glia is a very complex model and the results need to be interpreted with caution.

## Figures and Tables

**Figure 1 cells-11-04116-f001:**
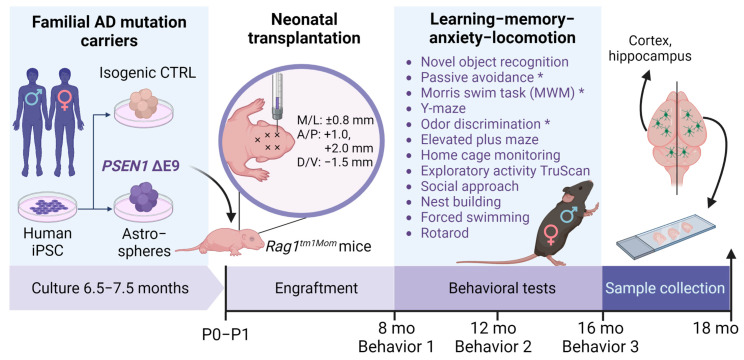
A schematic representation of the experimental setup, created with BioRender.com. *PSEN1* ΔE9 iPSCs obtained by reprogramming the patient-derived fibroblasts and isogenic CTRL iPSCs were differentiated into glial progenitors using the astrosphere method. Glial progenitor cells from 6.5–7.5 months-old astrospheres were dissociated and injected intraventricularly into newborn mouse pups. The mice then underwent a battery of behavioral tests at the ages of 8, 12, and 16 months. Brain samples were collected at the age of 18 months. Behavioral tests marked with * were performed only once at the age of 16 months. (The figure is created with BioRender.com; accessed on 16 December 2022).

**Figure 2 cells-11-04116-f002:**
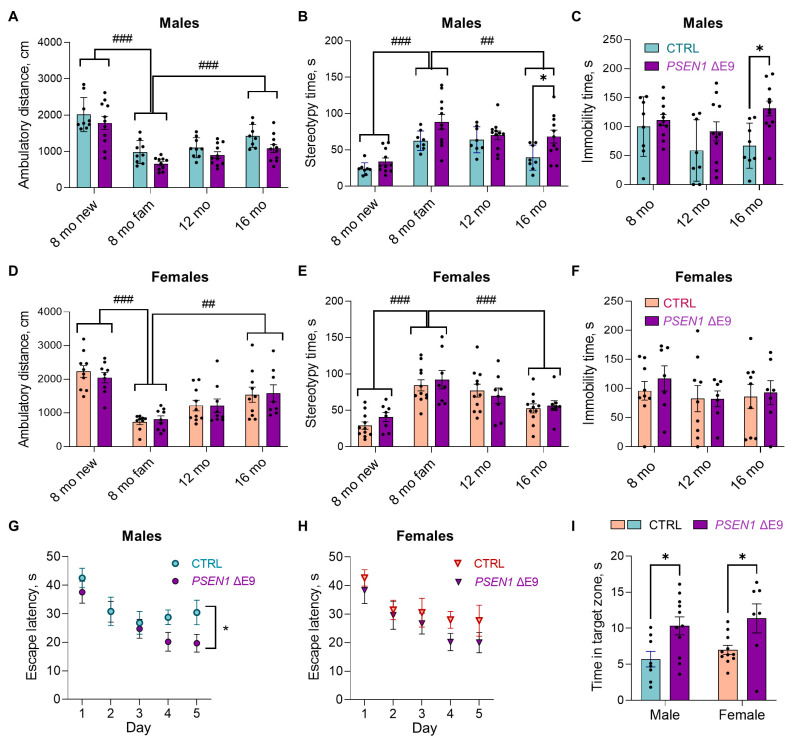
*PSEN1* ΔE9 mutation in human glial progenitors induced behavioral changes in 16-month-old mice. (**A**,**B**,**D**,**E**) Total distance traveled (cm) (**A**,**D**) and stereotypy time (s) (**B**,**E**) during a 10-min spontaneous exploration in the TruScan device at the ages of 8, 12, and 16 months in male (**A**,**B**) and female (**D**,**E**) mice. At the age of 8 months, the mice were exposed to the TruScan environment twice: first when it was a completely new environment (8-month new) and second when it was already a familiar environment (8-month fam). Purple bars indicate *PSEN1* ΔE9 glia-transplanted mice, blue bars indicate CTRL-transplanted males, and salmon CTRL-transplanted females; dots represent values obtained from individual animals. (**C**,**F**) Immobility (floating) time (s) recorded during the last 4 min of a 6-min forced swim test. (**G**,**H**) Escape latency (s) to the submerged platform in Morris swim task (watermaze). (**I**) Time spent at the former platform position in the 2nd probe trial of Morris swim task (s). The data in the graphs are shown as mean ± SEM; *n* = 8–11 per group; * *p* < 0.05 as compared to sex- and age-matched CTRL; ^##^ *p* < 0.01, ^###^ *p* < 0.001.

**Figure 3 cells-11-04116-f003:**
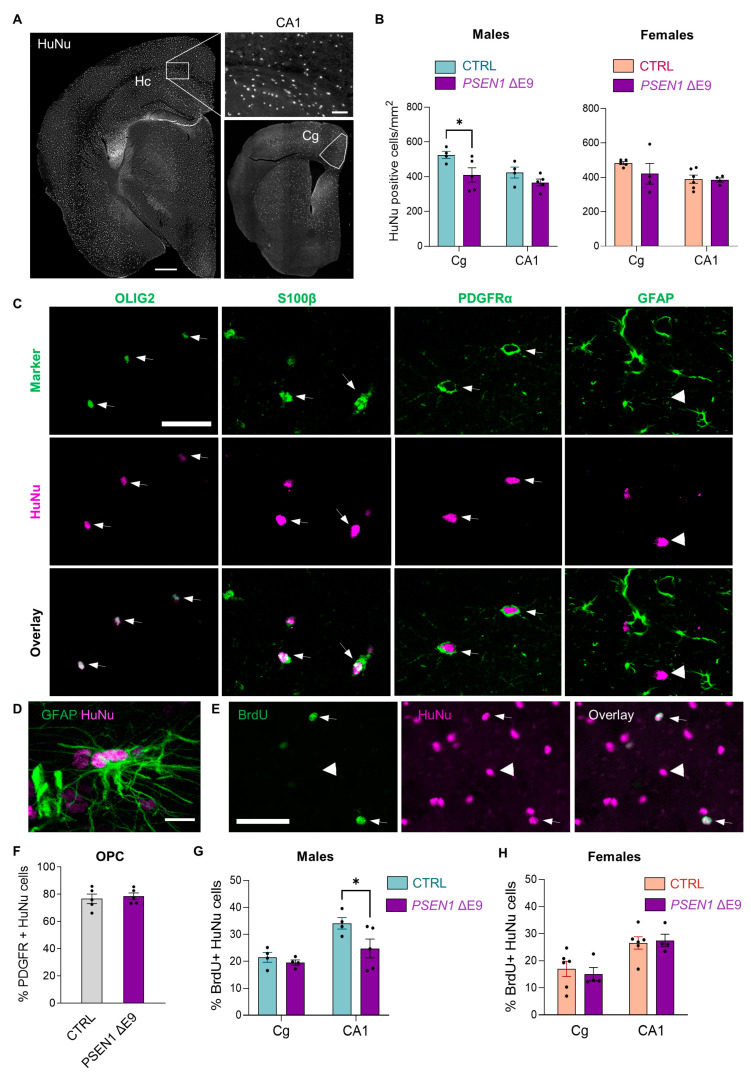
Grafted human progenitor cells survived for 18 months in the mouse brain and gave mostly to OLIG2/PDGFRα-expressing OPCs. (**A**) Representative images of HuNu staining in the mouse brain, including the hippocampus (Hc) and cingulate cortex (Cg) areas, scale bar 400 µm. HuNu-labeled cells in the CA1 area of the hippocampus are also shown at larger magnification, scale bar 50 µm. (**B**) The average density of HuNu-positive cells in Cg and CA1 areas. (**C**) Colocalization of HuNu with OLIG2, S100β, PDGFRα, and GFAP immunostaining in CA1 area, arrows, double-positive cells; arrowheads, GFAP-negative HuNu cells, scale bar 30 µm. (**D**) A representative image of HuNu colocalization with GFAP in the corpus callosum. (**E**) Colocalization of HuNu with BrdU; arrows, double-positive cells; arrowheads, BrdU-negative human cell; scale bar 50 µm. (**F**) Percentages of HuNu-positive cells expressing PDGFRα in the CA1 area; purple bars indicate *PSEN1* ΔE9 glia-transplanted mice, blue bars indicate CTRL-transplanted males, and salmon CTRL-transplanted females; dots represent values obtained from individual animals. (**G**,**H**) Percentages of HuNu-positive cells colocalizing with BrdU in Cg and CA1 areas. The data in the graphs are shown as mean ± SEM; *n* = 4–6 per group; * *p* < 0.05 as compared to sex-matched CTRL.

**Figure 4 cells-11-04116-f004:**
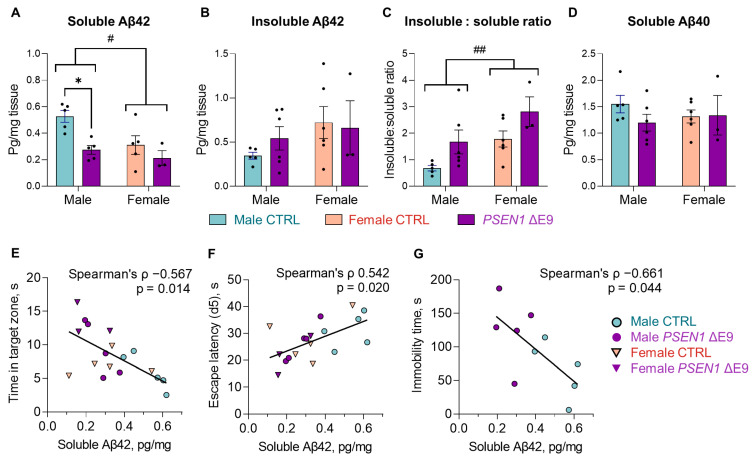
*PSEN1* ΔE9 mutant human cells convert a higher proportion of self-produced Aβ42 to insoluble form than the control cells. (**A**,**B**) The levels of TBS-soluble (**A**) and insoluble (guanidine fraction; (**B**) Aβ42 in the cortex expressed as pg/mg tissue. (**C**) The ratio of Aβ42 levels in guanidine fraction (insoluble) to TBS fraction (soluble). (**D**) The level of TBS-soluble Aβ40 in the cortex expressed as pg/mg tissue. Purple bars indicate *PSEN1* ΔE9 glia-transplanted mice, blue bars indicate CTRL-transplanted males, and salmon CTRL-transplanted females; dots represent values obtained from individual animals. The data in the graphs are shown as mean ± SEM; *n* = 3–6 per group; * *p* < 0.05 as compared to sex-matched CTRL; ^#^ *p* < 0.05; ^##^ *p* < 0.01. (**E**–**G**) Correlation plots of cortical soluble Aβ42 levels (pg/mg) with the time spent in the target zone (s) in the 2nd probe trial of Morris swim task (**E**), escape latency (s) on day 5 of training phase (**F**), and with the immobility time (s) in forced swim task (**G**). In plots E and F, both male and female data are included; in plot G, only the data from males.

**Figure 5 cells-11-04116-f005:**
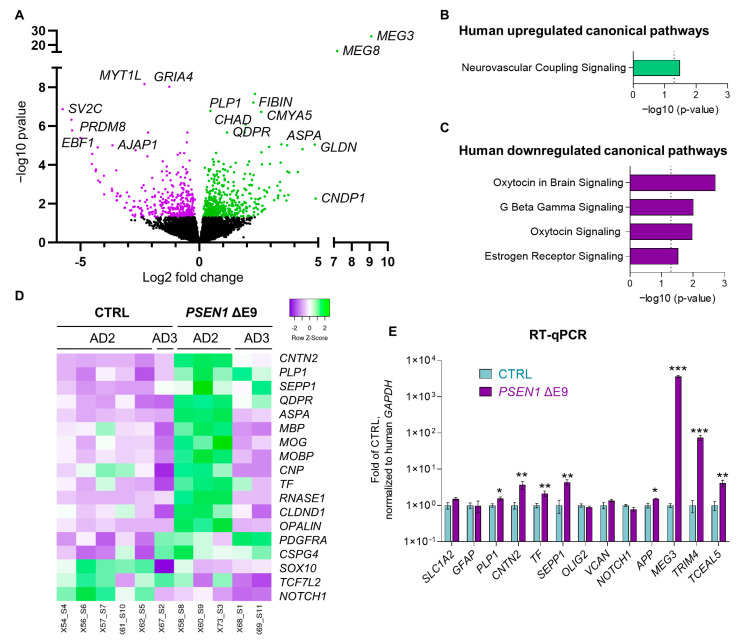
*PSEN1* ΔE9 mutant human glial cells exhibit an altered transcriptome. (**A**) A volcano plot showing differentially expressed genes in *PSEN1* ΔE9 mutant human cells in mouse hippocampi. (**B**,**C**) Ingenuity pathway analysis of human *PSEN1* ΔE9 vs. CTRL DEGs (**B**,**C**). Significantly upregulated pathways (**B**) indicate Z-score ≥ 1.99; significantly downregulated pathways (**C**) indicate Z-score ≤ −1.99; threshold of significance –log10 (*p*-value) = 1.3 (*p* = 0.05; dotted line). (**D**) The heatmap of human oligodendrocyte-associated genes showing the relative expression changes for each animal labeled by genotype (CTRL vs. *PSEN1* ΔE9) and parental iPSC line of the transplanted cells (AD2 vs. AD3), purple indicates low and green high expression level normalized to row average. (**E**) RT-qPCR data validating transcriptional changes in selected human genes normalized to human housekeeping gene *GAPDH,* shown as fold of expression in isogenic CTRL cells. Purple bars indicate *PSEN1* ΔE9 glia-transplanted mice, blue bars indicate CTRL-transplanted males. The data in the graphs are shown as mean ± SEM; *n* = 4–6 per group; * *p* < 0.05, ** *p* < 0.01, *** *p* < 0.001 as compared to CTRL.

**Figure 6 cells-11-04116-f006:**
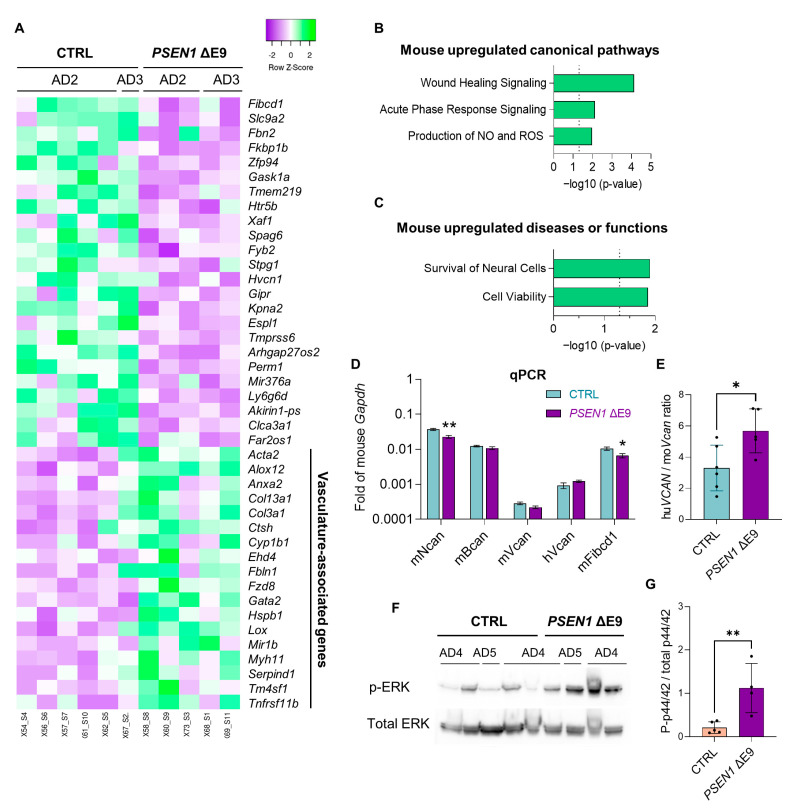
*PSEN1* ΔE9 mutant human glial cells induce changes in neighboring mouse cells in hippocampi. (**A**) A heatmap showing the relative expression changes in differentially expressed mouse genes labeled by genotype (CTRL vs. *PSEN1* ΔE9) and parental iPSC line of the transplanted cells (AD2 vs. AD3); purple indicates low and green high expression level normalized to row average. (**B**,**C**) Ingenuity pathway analysis of mouse *PSEN1* ΔE9 vs. CTRL DEGs; significantly upregulated pathways indicate Z-score ≥ 1.99; threshold of significance –log10 (*p*-value) = 1.3 (*p* = 0.05; dotted line). (**D**) RT-qPCR data showing relative expression of CSPG-related genes, all shown as fold of expression of mouse housekeeping gene *Gapdh.* (**E**) A relative expression of the human *VCAN* gene shown as the fold of mouse *Vcan* gene expression. (**F**,**G**) Representative immunoblot labeled by genotype (CTRL vs. *PSEN1* ΔE9) and parental iPSC line of the transplanted cells (AD2 vs. AD3) (**F**) and quantification analysis of phospho-ERK (Thr202/Tyr204) (**G**) in hippocampal lysates from female mice. Purple bars indicate *PSEN1* ΔE9 glia-transplanted mice, blue bars indicate CTRL-transplanted males, and salmon CTRL-transplanted females. The data in the graphs are shown as mean ± SEM; *n* = 4–6 per group; * *p* < 0.05, ** *p* < 0.01 as compared to CTRL.

**Figure 7 cells-11-04116-f007:**
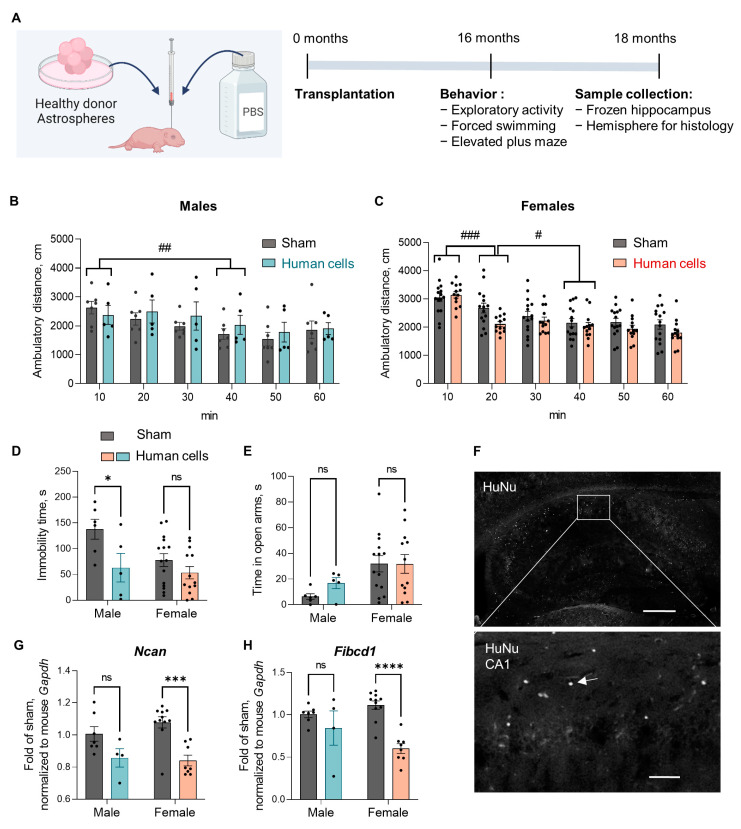
Human glia transplantation decreased immobility in the forced swim test in 16-month-old mice as compared to sham-operated controls. (**A**) A schematic representation of the experimental setup, created with BioRender.com (accessed on 15 December 2022); PBS, phosphate-buffered saline. (**B**,**C**) Total distance (cm) traveled by male (**B**) or female mice (**C**) in an open arena shown in 10-min blocks. (**D**) Immobility (floating) time (s) recorded during the last 4 min of a 6-min forced swim test. (**E**) Total time (s) spent in the open arms of the elevated plus maze. (**F**) A representative image of HuNu staining in the hippocampus, scale bar 400 µm. HuNu-labeled cells in the CA1 area of the hippocampus are also shown at larger magnification, scale bar 50 µm; arrow, HuNu-positive cell. (**G**,**H**) The relative mRNA expression level of mouse *Ncan* and *Fibcd1* shown as fold of mouse *Gapdh* in the hippocampus. Grey bars indicate sham-operated mice, blue bars indicate human glia-transplanted males, and salmon human glia-transplanted females. The data in the graphs are shown as mean ± SEM; *n* = 4–15 per group; * *p* < 0.05, *** *p* < 0.001, **** *p* < 0.0001 as compared to sham; ^#^ *p* < 0.05, ^##^ *p* < 0.01, ^###^ *p* < 0.001.

**Table 1 cells-11-04116-t001:** Summary of the human-derived iPSC lines used in this study.

Line	Sex	Age at Biopsy	*PSEN1* Genotype	*APOE* Genotype	Transplanted Mice Referred to in the Text as
**Batch I (main)**					
AD2	M	48	ΔE9/CTRL	ε3/ε3	*PSEN1* ΔE9
AD2-iso			corrected	ε3/ε3	CTRL
AD3	F	47	ΔE9/CTRL	ε3/ε3	*PSEN1* ΔE9
AD3-iso			corrected	ε3/ε3	CTRL
**Batch II**					
MADGIC 1	M	67	CTRL	ε3/ε3	Human cells
MADGIC 8	M	64	CTRL	ε3/ε3	Human cells

**Table 2 cells-11-04116-t002:** Summary of the transplantations.

iPSC Genotype	iPSC Line	Transplanted Pups Total	Males	Females
**Batch I (main)**				
*PSEN1* ΔE9	AD2	9	4	5
	AD3	13	8	5
Isogenic (CTRL)	AD2-iso	10	5	5
	AD3-iso	16	7	9
**Batch II**				
CTRL	MADGIC 1	9	3	6
	MADGIC 8	7	2	5
Sham		22	7	15

**Table 3 cells-11-04116-t003:** Summary of behavioral tests used in this study.

Behavioral Test	Outcome Measure
Home cage monitoring	Normal behavior
Exploratory activity TruScan	Locomotion, exploratory activity
Nest building	General health
Rotarod	Motor coordination
Social approach	Social behavior, cognition
Elevated plus maze	Anxiety
Forced swimming	Despair
Spontaneous alternation in a Y-maze	Exploratory behavior, cognitive function
Novel object recognition	Memory (perirhinal cortex)
Morris swim task (water maze)	Spatial learning
Odor discrimination	Memory (piriform cortex)
Passive avoidance	Learning, memory

**Table 4 cells-11-04116-t004:** Summary of the antibodies used in this study.

**Primary Antibodies**	**Host**	**Specificities**	**Supplier**	**Catalog #**	**Application**	**Dilution**	**RRID #**
HuNu	mouse	human nuclei	Merck	MAB4383	IHC	1:500	AB_827439
BrdU	rat	species independent	Abcam *	ab6326	IHC	1:400	AB_305426
S100β	rabbit	mouse, rat	Abcam	ab41548	IHC	1:5000	AB_956280
PDGFRα	rabbit	human	Cell Signaling Technology **	5241	IHC	1:300	AB_10692773
GFAP	rabbit	human, mouse	Agilent Dako ***	Z0334	IHC	1:500	AB_10013382
Olig2	rabbit	human, mouse	Merck	AB9610	IHC	1:1000	AB_570666
Aβ peptide	rabbit	human, mouse	Thermo Fisher Scientific	51–2700	IHC	1:1000	AB_2533902
p-AKT (Ser473)	rabbit	human, mouse	Cell Signaling Technology	4060	WB	1:1000	AB_2315049
AKT	rabbit	human, mouse	Cell Signaling Technology	4691	WB	1:2000	AB_915783
p-ERK (Thr202/Tyr204)	rabbit	human, mouse	Cell Signaling Technology	9101	WB	1:1000	AB_331646
ERK	rabbit	human, mouse	Cell Signaling Technology	9102	WB	1:1000	AB_330744
β actin	mouse	human, mouse	Merck	A5441	WB	1:5000	AB_476744
**Secondary Antibodies**	**Host**	**Specificities**	**Supplier**	**Catalog #**	**Application**	**Dilution**	**RRID #**
Alexa Fluor 555	donkey	mouse	Thermo Fisher Scientific	A31570	IHC	1:500	AB_2536180
Alexa Fluor 488	donkey	rat	Thermo Fisher Scientific	A21208	IHC	1:500	AB_2535794
Alexa Fluor 488	donkey	rabbit	Thermo Fisher Scientific	A21206	IHC	1:500	AB_2535792
Alexa Fluor 488	goat	rabbit	Thermo Fisher Scientific	A11008	IHC	1:800	AB_143165
Alexa Fluor 568	goat	mouse	Thermo Fisher Scientific	A11004	IHC	1:800	AB_2534072
Biotinylated	goat	mouse	Vector Laboratories	BA-9200	IHC	1:500	AB_2336171
Biotinylated	goat	rabbit	Vector Laboratories	BA-1000	IHC	1:500	AB_2313606
HRP	goat	rabbit	Thermo Fisher Scientific	A16096	WB	1:10000	AB_2534770
HRP	rabbit	mouse	Merck	A9044	WB	1:20000	AB_258431

RRID, research resource identifier; IHC, immunohistochemistry; WB, Western blotting; HRP, horseradish peroxidase. * Abcam, Cambridge, United Kingdom; ** Cell Signaling Technology, Danvers, MA, USA; *** Agilent, Santa Clara, CA, USA.

**Table 5 cells-11-04116-t005:** Summary of qPCR primers used in this study.

Target	Specificity	Supplier	Catalog #
*SLC1A2*	human	Thermo Fisher Scientific	Hs01102423_m1
*GFAP*	human	Thermo Fisher Scientific	Hs00909233_m1
*PLP1*	human	Thermo Fisher Scientific	Hs00166914_m1
*CNTN2*	human	Thermo Fisher Scientific	Hs00543989_m1
*TF*	human	Thermo Fisher Scientific	Hs00169070_m1
*SEPP1*	human	Thermo Fisher Scientific	Hs01032845_m1
*OLIG2*	human	Thermo Fisher Scientific	Hs00377820_m1
*VCAN*	human	Thermo Fisher Scientific	Hs00171642_m1
*NOTCH1*	human	Thermo Fisher Scientific	Hs01062014_m1
*APP*	human	Thermo Fisher Scientific	Hs00169098_m1
*MEG3*	human	Thermo Fisher Scientific	Hs00292028_m1
*TRIM4*	human	Thermo Fisher Scientific	Hs00982695_m1
*TCEAL5*	human	Thermo Fisher Scientific	Hs01383798_m1
*GAPDH*	human	Thermo Fisher Scientific	Hs99999905_m1
*Ncan*	mouse	Thermo Fisher Scientific	Mm00484007_m1
*Bcan*	mouse	Thermo Fisher Scientific	Mm00476090_m1
*Vcan*	mouse	Thermo Fisher Scientific	Mm01283063_m1
*Fibcd1*	mouse	Thermo Fisher Scientific	Mm00619124_m1
*Gapdh*	mouse	Thermo Fisher Scientific	Mm99999915_g1

## Data Availability

Appendix A show the full list of human (Appendix A) and mouse (Appendix A) differentially expressed genes with raw expression values. Appendix A show the list of significantly enriched pathways in human (Appendix A) and mouse (Appendix A) datasets identified by IPA software. The raw RNA-sequencing data isuploaded in full to the GEO database: GSE221027 (https://www.ncbi.nlm.nih.gov/geo/query/acc.cgi?acc=GSE221027 (accessed on 20 November 2022)). Appendix A show the data from all the behavioral tests conducted that were not included in the main part of the manuscript. Appendix A shows the behavioral data presented in Figure 2 as stratified by the parental iPSC line of the transplanted cells. Appendix A shows representative images of Aβ-positive aggregates in transplanted mouse brains and the quantification data. Appendix A shows raw Western blot images corresponding to Figure 6F.

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
