# Peer review of "Human PSEN1 Mutant Glia Improve Spatial Learning and Memory in Aged Mice"

_cells, 2022, doi:10.3390/cells11244116_

Round 1

Reviewer 1 Report

The present work entitled: “Human PSEN1 mutant glia improve spatial learning and memory in aged mice” by Henna Jäntti, Minna Oksanen and collaborators is well written and accessible to a broad audience. The authors did an outstanding work including males and females in their study and reporting separately the results from both. It is interesting to see that the phenotypic changes are prominent in males in the context of Alzheimer disease. The experimental settings are very well reported making the work reproducible. The results provide important clues about the involvement of glial cells in Alzheimer diseases and provides additional data about the transcriptomic changes induced in mice brain following the implantation of human cells. The results are supporting the authors claims and the figures are clear. I only have a few comments regarding this work:

Major:

Could the origin of the cells AD2 (M) and AD3 (F) explains the variability observed at the transcript levels in the male mice? This should be discussed more.

The RNA seqs were performed on the mouse samples. Further analysis on the mouse reads should be performed. At least providing heatmap or volcan plots of the differentially expresses genes and pathway analysis as these would provide important information about the model. It would be interesting to include in the results how the human cells affect the mouse cells and discuss the findings.

minor:

Line 47: a reference could be added to the epidemiology

Line 48: the name of protein encoded by PSEN1, presenilin 1, could be included.

Line 63: in vitro should be italicized

Line 72: in vivo should be italicized

Fig5J : please provide the AD information above the blots. Could it participate in the observed variability?

Fig4G: the caption in the graph seems to have been shifted.

Author Response

Could the origin of the cells AD2 (M) and AD3 (F) explains the variability observed at the transcript levels in the male mice? This should be discussed more.

The RNA seqs were performed on the mouse samples. Further analysis on the mouse reads should be performed. At least providing heatmap or volcan plots of the differentially expresses genes and pathway analysis as these would provide important information about the model. It would be interesting to include in the results how the human cells affect the mouse cells and discuss the findings.

Dear reviewer,

Thank you very much for your suggestions! Unfortunately, we did not include sham-operated mice in the transcriptomics dataset. Therefore, our data do not allow to make conclusions about how human glia affect mouse cells in general. We could investigate this aspect in our next paper.

To make things more clear we have decided to divide Figure 5 into 2 Figures. The new Figure 5 is smaller and includes only human data (please see the Word file attached). The new Figure 6 now shows the mouse gene data from the same transcriptomics dataset). We have added here the heatmap of the most interesting downregulated and upregulated mouse genes as Figure 6A. We also have included an explanation why we could not dissect out significantly downregulated pathways: "Out of 67 down-regulated genes, 30 have unknown function, 6 are poorly characterized lnc RNAs, and 7 exhibited a very low expression level. The relative expression of the rest 24 downregulated genes and of vasculature-associated upregulated genes is shown in Figure 6A" (line 672-674, Word file attached). We have also added the sentence: "The sex of the parental line (male AD2 vs. female AD3) may have contributed to the observed variation" (line 660-661, Word file attached).

minor:

Line 47: a reference could be added to the epidemiology. Corrected here 50 % to 30% and added the reference (see Word file attached).

Line 48: the name of protein encoded by PSEN1, presenilin 1, could be included - Corrected (see Word file attached).

Line 63: in vitro should be italicized - Corrected (see Word file attached).

Line 72: in vivo should be italicized - Corrected (see Word file attached).

Fig5J : please provide the AD information above the blots. Could it participate in the observed variability? The blots are now shown as Figure 6F and the AD line information is provided. We have also added the sentence "The within-group variation could not be explained by the origin of the parental iPSC line" (line 723-724, Word file attached).

Fig4G: the caption in the graph seems to have been shifted. The caption is corrected.

Reviewer 2 Report

A very interesting study has been conducted . The article is well written, the research is very relevant and will be interesting to many readers not only from the neurodegenerative field.

Author Response

Dear reviewer,

Thank you very much for your encouraging comments! Due to some suggestions coming from Reviewer 1, we have made a few minor changes in the manuscript:

1. To make things more clear we have decided to divide Figure 5 into 2 Figures. The new Figure 5 is smaller and includes only human data (please see the Word file attached). The new Figure 6 now shows the mouse gene data from the same transcriptomics dataset). We have added here the heatmap of the most interesting downregulated and upregulated mouse genes as Figure 6A. We also have included an explanation why we could not dissect out significantly downregulated pathways: "Out of 67 down-regulated genes, 30 have unknown function, 6 are poorly characterized lnc RNAs, and 7 exhibited a very low expression level. The relative expression of the rest 24 downregulated genes and of vasculature-associated upregulated genes is shown in Figure 6A" (line 672-674, Word file attached). We have also added the sentence: "The sex of the parental line (male AD2 vs. female AD3) may have contributed to the observed variation" (line 660-661, Word file attached).

2. Line 47: Corrected here 50 % to 30% and added the reference (see Word file attached).

3. Line 48: added the name of the protein here: "Deletion of exon 9 in the human PSEN1 gene encoding for presenilin 1 (PSEN1 ΔE9) causes an early-onset form of AD characterized by the accumulation of amyloid plaques, neurofibrillary tangles, and cerebral amyloid angiopathy" (see Word file attached).

4. Line 63: in vitro is italicized  (see Word file attached).

5. Line 72: in vivo is italicized (see Word file attached).

6The Western blots for p-ERK are now shown as Figure 6F and the AD line information is provided. We have also added the sentence "The within-group variation could not be explained by the origin of the parental iPSC line" (line 723-724, Word file attached).

7. Fig4G: the caption in the graph is corrected.

Reviewer 3 Report

The authors Jäntti et al., elucidate an important issue about investigating the effect of the PSEN1 ΔE9 mutation on human glial phenotype in vivo, on mouse behavior, cognition, and Aβ deposition.

i) Aged PSEN1ΔE9-transplanted mice showed improved spatial learning and memory

ii) PSEN1 ΔE9-transplanted mice had also lower cortical levels of soluble Aβ42

iii) PSEN1 ΔE9 mutation was associated with altered regulation of chondroitin sulfate proteoglycan  signaling in the mouse hippocampus.

iv) lastly, the PSEN1 ΔE9-transplanted mice might be affected by an impairment of neurovascular coupling.

The manuscript is a well-written research paper and the length of the paper is commensurate with the message. The multilevel approaches permit a rigorous experimental analysis from molecular, cellular, and gene expression up to behavioral and cognitive tasks. The discussion focused on all showed experimental steps. The literature cited is of important relevance. This work illustrates a solid experimental paradigm for the transplantation of human glia. For the mentioned reasons, the manuscript may be accepted for publication without revision.

Author Response

(The authors gave the same response as above.)
